# Investigating Layer Importance in Large Language Models

**Yang Zhang**[1]    **Yanfei Dong**[1,2]    **Kenji Kawaguchi**[1]
[1]National University of Singapore    [2]PayPal
{yangzhang, dyanfei, kenji}@comp.nus.edu.sg

## Abstract

Large language models (LLMs) have gained increasing attention due to their prominent ability to understand and process texts. Nevertheless, LLMs largely remain opaque. The lack of understanding of LLMs has obstructed the deployment in safety-critical scenarios and hindered the development of better models. In this study, we advance the understanding of LLM by investigating the significance of individual layers in LLMs. We propose an efficient sampling method to faithfully evaluate the importance of layers using Shapley values, a widely used explanation framework in feature attribution and data valuation. In addition, we conduct layer ablation experiments to assess the performance degradation resulting from the exclusion of specific layers. Our findings reveal the existence of cornerstone layers, wherein certain early layers can exhibit a dominant contribution over others. Removing one cornerstone layer leads to a drastic collapse of the model performance, often reducing it to random guessing. Conversely, removing non-cornerstone layers results in only marginal performance changes. This study identifies cornerstone layers in LLMs and underscores their critical role for future research.

## 1 Introduction

The rapid advancement of large language models (LLMs) has revolutionized natural language processing, enabling unprecedented capabilities in text generation, translation, and comprehension tasks c̃itewei2022chain, hu2021lora, rafailov2024direct, ouyang2022training. These models, exemplified by architectures such as GPT-3 (Brown et al., 2020), Llama (Touvron et al., 2023a,b), and Bloom (Workshop et al., 2022), rely on transformer-based neural networks with numerous layers (Vaswani et al., 2017). Despite their successes, LLMs suffer from issues such as hallucinations, biases, and unstable reasoning abilities (Hendrycks et al., 2020; Bolukbasi et al., 2016;

Bender et al., 2021; Garg et al., 2018). Regardless of the effort to mitigate these issues (Cao et al., 2018; Huang et al., 2023; Dathathri et al., 2019; Kaneko and Bollegala, 2021), they remain unsolved nowadays, hindering the deployment of LLMs in more safety-critical domains. When a neural network makes errors or underperforms, it is valuable to identify the specific part of the model responsible for these issues. Therefore, understanding the inner workings of neural networks and recognizing the role of individual components is key to addressing the challenges associated with LLMs.

In this paper, we advance the understanding of LLMs by investigating the importance of individual layers in LLMs across multiple tasks. To quantify the contribution of each layer to the overall model performance, we extend the Shapley value framework (Lundberg and Lee, 2017; Ghorbani and Zou, 2020), originally from cooperative game theory, to layers in LLMs. We employ an efficient sampling method to estimate layer importance within a practical runtime. To further analyze the impact of the key layers characterized by high Shapley values, we perform layer ablation to observe a specific layer's impact on performance.

Our study reveals that certain early layers in LLMs, which we term *cornerstone layers*, play a dominant role in influencing the model's performance. Notably, removing one of these cornerstone layers can cause a significant performance drop, reducing the model performance to near random guessing. In contrast, removing other layers typically results in only marginal performance degradation. We hypothesize that these cornerstone layers handle some fundamental tasks in LLMs and hope this discovery inspires future studies on understanding the role of cornerstone layers.

**Our contribution:** (1) We propose an efficient sampling method based on the proximity of LLM layers to estimate layer Shapley values. (2) We investigate the importance of layers in LLMs using

layer Shapley with layer ablation. Our method complements the traditional model explanation method with a mechanistic interpretability perspective. (3) We identify cornerstone layers in LLMs. A cornerstone layer has distinct behavior compared to other layers. It has a major contribution across many tasks, and its absence leads to the collapse of model performance. (4) We also examine the behavior of cornerstone layers across different models and tasks. Our findings demonstrate the universal importance of these layers across various tasks, while also revealing that cornerstone layers contribute differently depending on the model. (5) We analyze our findings and provide two possible hypotheses for the observed model behavior.

## 2 Related Work

There is a significant body of research focused on interpreting and understanding large language models (LLMs). This section provides an overview of some key approaches.

**Analysing parts of LLMs:** Shim et al. (2022) analyze the contributions of various components in LLMs and their impacts on performance. Gromov et al. (2024) investigate the role of deep layers in LLMs through layer pruning. Michel et al. (2019) explore the redundancy of attention heads, showing that many heads can be pruned without significant performance loss. Clark et al. (2019b) study the behavior of individual attention heads in BERT, revealing their distinct roles in capturing linguistic features.

**Model probing:** Probing techniques are widely used to analyze the internal representations of LLMs: Tenney et al. (2019) use probing tasks to examine what linguistic information BERT captures, finding that different layers encode different types of linguistic features. Tenney et al. (2018) introduce a suite of probes to analyze the representations learned by contextualized word embeddings, identifying how syntactic and semantic information is distributed across layers.

**Mechanistic interpretability:** Some research views the inner workings of LLMs as circuits: Pal et al. (2023) conceptualize LLMs as computational circuits, mapping out how information flows through the network. Meng et al. (2022) focus on locating and understanding functional circuits within LLMs, providing insights into how factual knowledge is stored in LLMs.

**Study of intermediate representation:** Understanding the intermediate representations within LLMs is another critical area of study: Sun et al. (2024) analyze the intermediate layers of LLMs, exploring how these representations evolve across the network and contribute to final predictions. Bricken et al. (2023) investigate the nature of intermediate representations and their roles in encoding syntactic and semantic information.

## 3 Preliminaries

### 3.1 Layers in LLMs

Recent LLMs primarily adopt a decoder-only architecture. Typically, an LLM begins with an embedding layer $E$, succeeded by $L$ transformer decoder layers $H_1, H_2, \ldots, H_L$, and ends with a head layer $C$ that predicts the probability of each token in the vocabulary $\mathcal{V}$. Each decoder layer $H_l$ consists of an attention layer and a feed-forward network (FFN) layer. Given an input prompt $\mathbf{x}$ of length $N$ and a vocabulary $\mathcal{V}$, where $\mathbf{x} \in |\mathcal{V}|^N$, the LLM first maps $\mathbf{x}$ into a hidden space, resulting in

$$\mathbf{h}_0 = E(\mathbf{x}),$$

where $\mathbf{h}_0 \in R^{N \times d}$. The hidden state $\mathbf{h}_0$ is then processed sequentially through the decoder layers:

$$\begin{aligned}
\boldsymbol{h}_l' &= \mathsf{Attn}_l(\boldsymbol{h}_{l-1}) + \boldsymbol{h}_{l-1} \quad \text{for } l = 1, 2, \ldots, L, \\
\boldsymbol{h}_l &= \mathsf{FFN}_l(\boldsymbol{h}_l') + \boldsymbol{h}_l' \quad \text{for } l = 1, 2, \ldots, L.
\end{aligned} \tag{1}$$

Lastly, the head layer $C$ predicts the logits

$$C(\boldsymbol{h}_L) = [\boldsymbol{z}^{(1)}, \boldsymbol{z}^{(2)}, \ldots, \boldsymbol{z}^{(N)}],$$

where $\boldsymbol{z}^{(i)} \in R^{|\mathcal{V}|}$ represents the predicted logits for the $(i+1)$-th output token.

### 3.2 Shapley Value

Shapley values, rooted in cooperative game theory, have become a powerful tool in the realm of explainable artificial intelligence (XAI), providing insights into the contributions of individual features within complex models. Originally formulated by Lloyd Shapley in 1953 (Shapley, 1952), Shapley values offer a systematic and fair allocation of payoffs to players based on their contributions to the total gain of a coalition, making them an essential method in understanding the role of each participant.

In the context of cooperative games, the Shapley value for a player represents the player's average

marginal contribution across all possible coalitions. This concept ensures that each player's influence is assessed comprehensively, considering every possible combination of players. Formally, for a set $N$ of $n$ players, the Shapley value $\phi_i$ for player $i$ is defined as:

$$\phi_i(v) = \sum_{S \subseteq N \setminus \{i\}} C \cdot [v(S \cup \{i\}) - v(S)],$$

where $v(S)$ represents the value of the coalition $S$, and $C$ is the combinatorial factor given by:

$$C = \frac{|S|!(n - |S| - 1)!}{n!}.$$

This formulation considers all permutations of players, ensuring that each player's contribution is fairly evaluated by analyzing every possible coalition they could potentially join.

Calculating Shapley values involves considering all subsets of players, which makes the computation particularly challenging as the number of players increases. For a game with $n$ players, the Shapley value requires an evaluation of $2^{n-1}$ possible subsets, leading to computational complexities that grow exponentially with $n$. Despite the challenges in computation, the Shapley value remains a cornerstone in XAI, particularly in attributing the contributions of different features in machine learning models. By fairly distributing credit among features, Shapley values enable a deeper understanding of model behavior, supporting transparency and trust in AI systems.

## 4 Estimating Layer Shapley

Prior works usually calculate shapley values between different input features or different data points. In a nutshell, Shapley values are usually applied to data, not models. Nevertheless, in this work, we adopt the well-established Shapley value framework to measure the contribution of a layer to the model performance. For a model with a sequential structure, we can construct its mathematical form in nested functions:

$$f(x) = f_N(f_{N-1}(...f_1(x))),$$

where $N$ is the number of layers in this model. Hence, we consider each layer $f_i$ as a player in the cooperative game. Specifically, we choose each individual attention and FFN layer as a player in LLMs and the model performance on a predefined task as the game outcome.

One major drawback of calculating Shapley values is the enormous number of required samples. To calculate Shapley values for N players precisely, one needs to sample $2^N$ times, which is computationally not feasible for LLMs. Therefore, we aim for a reasonable estimation of layer Shapley with efficient sampling that exploits the structure of LLMs and achieves orders of magnitude speed-ups, which we explain below.

**Early truncation:** As discussed in prior work that estimates Shapley values (Ghorbani and Zou, 2020), the model performance degrades to random guessing for cases where many layers are removed. Consequently, the value difference will be

$$|v(S \cup \{i\}) - v(S)| < \epsilon,$$

where $\epsilon$ is a small real number close to zero, since both $v(S \cup \{i\})$ and $v(S)$ are essentially random guesses. To exploit this, we limit our sampling to scenarios where layers are removed up to a certain level. Formally, we apply the constraint that $|S| > N_{lim}$, where $N_{lim}$ is a hyperparameter that defines the maximal layer perturbation level. This approach results in an overestimation of the layer Shapley values, as many near-zero contributions are excluded from the sampling process. However, the relative ordering of the Shapley values remains accurate.

**Neighborhood sampling:** Besides early truncation, we leverage the sequential structure of the model to perform efficient sampling. Each layer primarily influences its immediate subsequent layers and is influenced by its immediate preceding layers. Consequently, interactions between distant layers are weaker than those between closely positioned layers. To capture meaningful interactions with fewer samples, we implement neighborhood sampling that only samples subsequent layers for Shapley value estimation. Formally, a set $S$ of $n$ elements under neighborhood sampling has the form

$$S = \{a, a + 1, \ldots, a + (n-1)\},$$

where $a$ is an offset value.

**Complexity analysis:** By combining both early truncation and neighborhood sampling, we reduce the number of samples from $2^N$ to $\frac{(N+N_{min})(N-N_{min})}{2}$, where $N$ is the total number of layers and $N_{min}$ is the minimal remaining layers defined by us. In our experiments, we remove

maximally 4 layers from the model during the layer Shapley value estimation.

# 5 Mechanistic Interpretation via Layer-wise Ablation

One limitation of the Shapley value is that, on its own, it does not provide a concrete understanding of how the model's performance degrades when layers are removed. Aside from using Shapley values to quantify layer contribution considering layer-wise interactions, it is also important to understand the functional significance of individual layers. Specifically, we perform layer ablation for this endeavor. Layer ablation involves selectively removing a target layer from the model and observing the resulting impact on performance across various tasks. This approach helps us isolate and evaluate the unique contribution of that specific layer, independent of others.

In conventional LLM architectures, skip connections are employed in each layer, as discussed in Section 3.1 and Equation 1. Skip connections, which bypass one or more layers, allow information to be transferred directly from one layer to another non-adjacent layer. Hence, it is possible to remove a layer without entirely disrupting the flow of information through the network. Without skip connections, the removal of a layer would likely result in a complete breakdown of the model, as the information flow would be interrupted. Therefore, we perform layer ablation by removing one layer while keeping the skip connection around the removed layer to maintain the information flow within the model. For a module with skip connection in the form of:

$$f_{\text{module with skip}}(x) = f_{\text{module}}(x) + x.$$

Removing this module results in an ablated structure in the form of:

$$f_{\text{ablated}}(x) = x.$$

Here, the effect of the module is completely removed, but the information processed by previous layers can still pass to subsequent layers. An illustration of the single-layer ablation is shown in Figure 1.

Layer ablation complements Shapley values by providing a mechanistic perspective on the contribution of each layer. While Shapley values offer a mathematical framework to understand the importance of each layer in the context of all possible

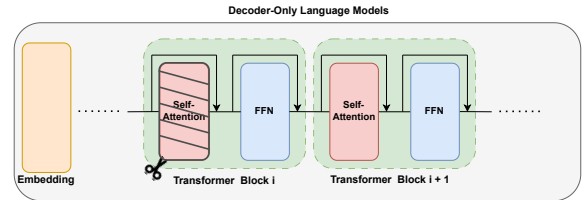

Figure 1: **Illustration of single-layer ablation**. A layer is ablated by removing the block while keeping the skip connection across the layer. We choose to ablate layers we used for layer Shapley calculation, that are, attention layers and FFN layers. For Mixtral 8x7B, we ablate attention layers and MoE layers. More details can be found in Section 5.

layer combinations, ablation experiments give us a direct way to observe the functional impact of a layer's removal. By combining both methods, we gain a comprehensive understanding of layer importance—Shapley values quantify the contribution in terms of interactions, while ablation highlights the practical significance of each layer in maintaining model performance.

# 6 Experiments

## 6.1 Experimental Setup

We evaluate the models on various datasets to ensure a comprehensive analysis of a wide spectrum of language understanding and reasoning tasks. In our study, we utilize three recent large language models to assess the impact of individual layers: LLaMA3-8B, LLaMA3-70B, and Mixtral-8x7B (Touvron et al., 2023b,a; Jiang et al., 2024). **LLaMA3-8B** contains 8 billion parameters, making it a mid-sized model suitable for a range of NLP tasks. **LLaMA3-70B** have 70 billion parameters and are significantly larger than LLaMA3-8B. This model is expected to capture more complex language patterns and dependencies. **Mixtral-8x7B** replaces FFN layers with Mixture-of-Expert (MoE) layers, each containing 8 experts. The ensemble approach aims to combine the strengths of multiple models to achieve superior performance.

We perform our experiment on 6 datasets ranging from simple to hard tasks. **BoolQ** (Clark et al., 2019a) is a reading comprehension dataset consisting of questions that can be answered with "yes" or "no" based on a given passage. **ARC-Easy and ARC-Challenge** (Clark et al., 2018) are part of the AI2 Reasoning Challenge (ARC), which provides multiple-choice questions derived from science exams. **PIQA** (Bisk et al., 2020) as-

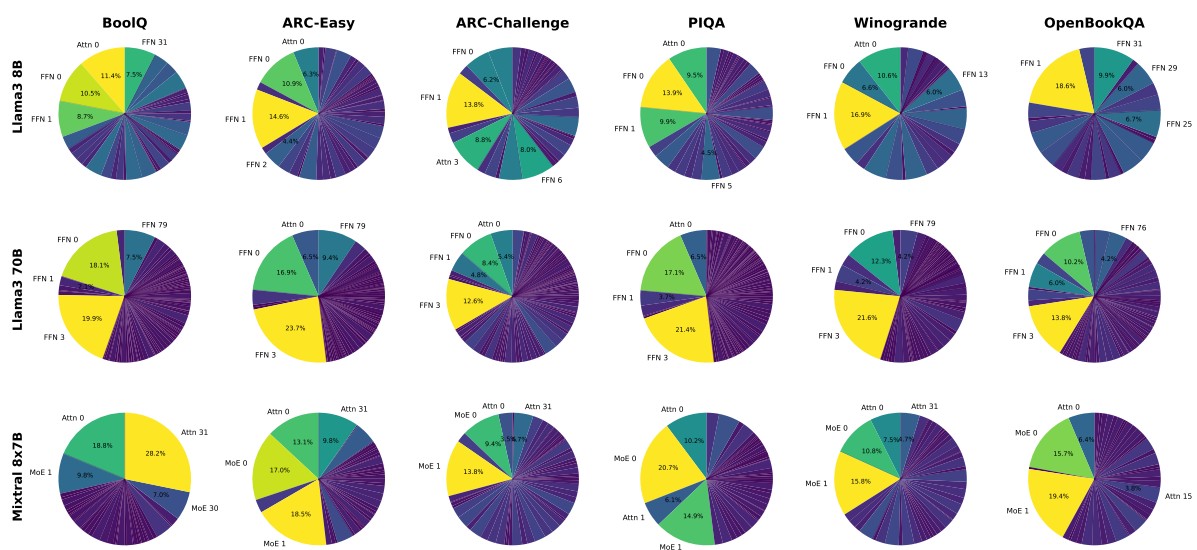

Figure 2: **Proportion of estimated layer Shapley values for each layer**. We calculate the proportion of Shapley values for each layer relative to all layers in the model. The layers in the pie chart are arranged in ascending order according to their proximity to the model input, moving in an anti-clockwise direction starting from the top of the chart. The top 4 most contributing layers are captioned. Across all three models (rows) and six tasks (columns), we observe a disproportionately high contribution from a few layers, typically early layers. Additionally, these important layers account for a significant portion of the overall layer importance. For example, in Llama3 70B, the top 4 layers contribute 47.6% to model performance, as indicated by Shapley values. More discussion in Section 6.2. Attn refers to attention layers, FFN refers to fully connected layers, and MoE refers to Mixture-of-Expert layers.

|  | Top 3 Layers | Other Layers |
| --- | --- | --- |
| Llama3 8B | 29.1% | 70.9% |
| Llama3 70B | 37.0% | 63.0% |
| Mixtral 8x7B | 37.5% | 62.5% |

Table 1: **Proportion of Shapley values summarized in two groups**. The top 3 layers with the highest Shapley values account for 30% of the total Shapley value. In larger models such as Llama3 70B and Mixtral 8x7B, the proportion of Shapley values attributed to the top three layers is even higher compared to smaller models.

|  | Cornerstone Layers |
| --- | --- |
| Llama3 8B | Attn 0, FFN 0, FFN 1 |
| Llama3 70B | Attn 0, FFN 0, FFN 3 |
| Mixtral 8x7B | Attn 0, MoE 0, MoE 1 |

Table 2: **Identified cornerstone layers**. These layers exhibit disproportionately high Shapley values compared to other layers across various tasks.

sesses the model's understanding of physical commonsense by presenting multiple-choice questions about everyday situations and interactions. **Winogrande** (Sakaguchi et al., 2019) includes sentence pairs with a pronoun that needs to be correctly resolved based on the Winograd Schema Challenge. **OpenBookQA** (Mihaylov et al., 2018) comprises questions that require knowledge from elementary science topics, testing the model's ability to combine factual knowledge with reasoning skills.

## 6.2 Shapley Value Result

This section shows results of estimated Shapley values. Figure 2 shows the proportion of estimated Shapley values (bar plot in Figure 6 in Appendix).

**Are there critical layers?** According to Figure 2 and Table 1, we observe a clear phenomenon that several layers contribute significantly to the model performance across all tasks. By grouping the top three layers with the highest Shapley values, we observe that they can take 29% to 37% of the total contribution on average across various tasks. In addition, models with Mixture-of-Expert layers, such as Mixtral-8x7B, and models with FFN layers, such as Llama models, share similar findings. Overall, we observe that several early layers possess a significantly higher contribution than other layers. On larger models such as Llama3-70B and Mixtral-8x7B models, the contribution distribution between layers is more unbalanced than a smaller Llama3-8B. As the layers with the most significant impact on model performance are typically the initial layers, we term them *cornerstone layers*.

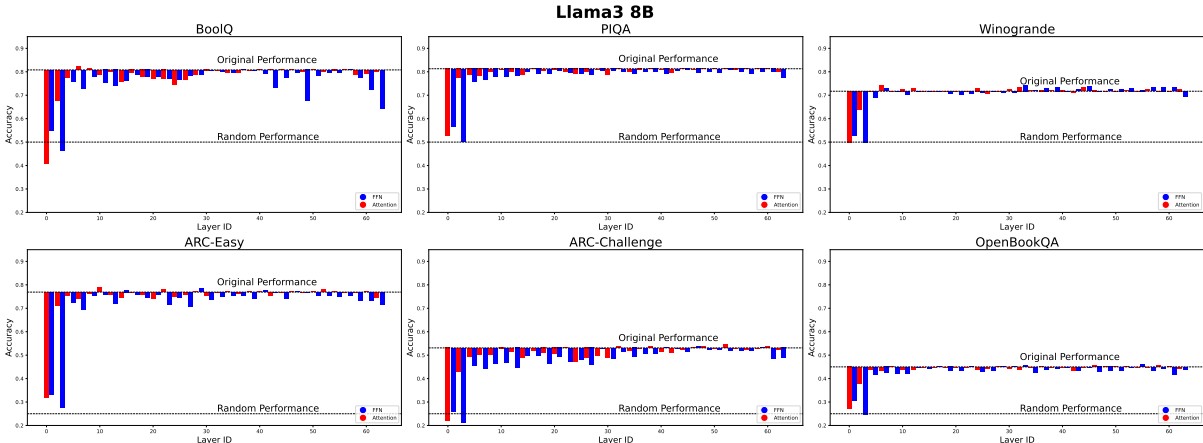

Figure 3: **Layer ablation result of Llama3 8B**. X-axis shows the layer ID of the removed layer. Y-axis shows the accuracy after this layer is removed. Attention layers are colored in red, while FFN layers are colored in blue. Removing one cornerstone layer can cause the model performance to immediately drop to random guesses. More discussion in Section 6.3.

**Where are critical layers located?** We observe that all models have cornerstone layers positioned in similar places. For Llama3-8B, we identify cornerstone layers to be the attention layer 0, the FFN layer 0, and the FFN layer 1. For Llama3-70B, cornerstone layers are the attention layer 0, the FFN layer 0, and the FFN layer 3. For Llama3-70B, cornerstone layers are the attention layer 0, the MoE layer 0, and the MoE layer 3. Table 2 shows a summary of cornerstone layers. The similar location of cornerstone layers suggests a similar processing flow among models.

**Are cornerstone layers more important in larger models?** In our analysis in Table 1, we have an additional observation that in larger models, the concentration of Shapley importance becomes even more pronounced, with the top three layers accounting for an even greater proportion of the total Shapley values compared to smaller models. This suggests that as models scale in size, the distribution of importance among layers becomes more uneven, with a few layers playing a disproportionately larger role in driving the model's overall effectiveness. Understanding this distribution is crucial for optimizing model architecture and improving interpretability, as it underscores the pivotal role of these key layers in the functioning of the model.

### 6.3 Layer Ablation Result

To complement insights acquired from layer Shapley studies and observe the practical effects of altering the model's architecture, we conduct layer ablation experiments. This dual approach allows us

to cross-validate our findings and formulate more robust hypotheses about the specific functions of cornerstone and non-cornerstone layers. Figure 3, Figure 4, and Figure 5 show the model performance after ablating one layer for Llama3 8B, Llama3 70B, and Mixtral 8x7B, respectively.

**How important are cornerstone layers?** According to Figure 3, Figure 4, and Figure 5, removing one layer with a high Shapley value causes the performance of the model to collapse and produce random guesses, while removing one from other layers only results in minor performance degradation, indicating their lesser importance. These cornerstone layers likely carry unique functionalities within the model, with their outputs serving as critical foundations for all subsequent layers.

**How unimportant are non-cornerstone layers?** Based on our results in Table 4, we find that non-cornerstone layers are less critical to the model's performance. This is evident from the small Shapley values of non-cornerstone layers as well as the minimal performance drop observed when a non-cornerstone layer is removed, suggesting that these layers play a less significant role compared to the cornerstone layers in the overall functioning of the model. Nevertheless, these layers are not entirely unimportant. According to our Shapley value and layer ablation experiments, they have small but non-zero contributions to the model.

**Are layers in MoE architecture better learned?** Intriguingly, the Mixtral-8x7B model is less reliant on cornerstone layers. According to Figure 5, ab-

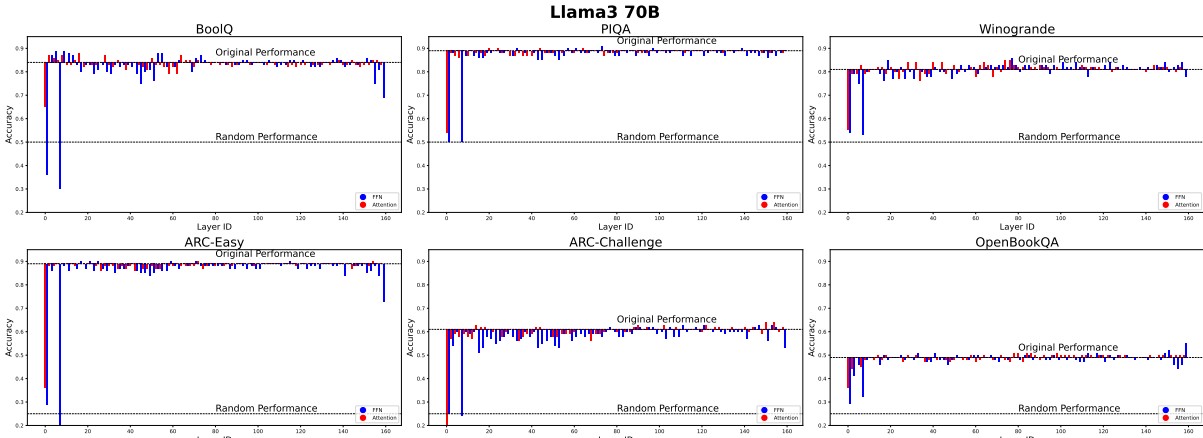

Figure 4: **Layer ablation result of Llama3 70B**. X-axis shows the layer ID of the removed layer. Y-axis shows the accuracy after this layer is removed. Attention layers are colored in red, while FFN layers are colored in blue. Similar to Llama3 8B, removing a single cornerstone layer causes the model's performance to degrade to the level of random guessing. More discussion in Section 6.3.

|  | C Layers | NC Layers |
|---|---|---|
| Llama3 8B | 29.3% | 1.6% |
| Llama3 70B | 36.7% | 0.9% |
| Mixtral 8x7B | 23.5% | 1.3% |

Table 3: **Performance drop after single-layer ablation averaged over tasks and layers**. Removing one cornerstone layer usually results in model collapse to random guessing, while removing one non-cornerstone layer causes minimal performance degradation.

lating these layers results in a smaller performance drop compared to Llama models. In five out of six tasks, Mixtral-8x7B maintains certain performance instead of dropping to random guessing when a cornerstone layer is removed. One likely explanation is that MoE layers provide more regularization through sparse activation of experts. This mechanism likely helps the model avoid over-relying on any single MoE layer.

## 6.4 Interpretation of Findings

In this section, we integrate the findings from Section 6.2 and Section 6.3 to hypothesize about the roles of cornerstone and non-cornerstone layers in the model. We observe that cornerstone layers are typically located at the beginning of an LLM and that removing these layers often causes the performance of the model to collapse to random guessing. In contrast, removing other layers results in only marginal performance changes. Based on these observations, we propose the following hypothesis:

**Hypothesis 1.** *Cornerstone layers are primarily*

|  | Lla. 8B | Lla. 70B | Mix. 8x7B |
|---|---|---|---|
| BoolQ | 2.8% | 1.1% | 2.1% |
| PiQA | 1.5% | 0.8% | 1.0% |
| WG | 0.4% | 0.6% | 1.4% |
| ARC-E | 1.6% | 1.1% | 1.2% |
| ARC-C | 2.6% | 1.6% | 2.1% |
| OBQA | 0.9% | 0.6% | 0.4% |

Table 4: **Performance drop after single-layer ablation averaged over non-cornerstone layers across tasks and models**. Removing one non-cornerstone layer has a neglectable effect on model performance on all tasks and models we used. WG: WinoGrande, OBQA: OpenbookQA, Lla.: Llama3, Mix.: Mixtral.

*responsible for processing the initial input embeddings, establishing the foundational outputs upon which every subsequent layer operates.*

For non-cornerstone layers, our results indicate that while their individual contributions are small, they are not insignificant. Their collective contribution can be substantial. Therefore, we propose the following hypothesis for non-cornerstone layers:

**Hypothesis 2.** *Non-cornerstone layers collaborate to process information, with their functionalities potentially overlapping.*

While our hypotheses are grounded in the findings from our analyses, we do not claim them to be definitive conclusions. Instead, we present these hypotheses to highlight the intriguing phenomena observed in our study, emphasizing the need for further investigation and validation. We encourage the research community to rigorously test these

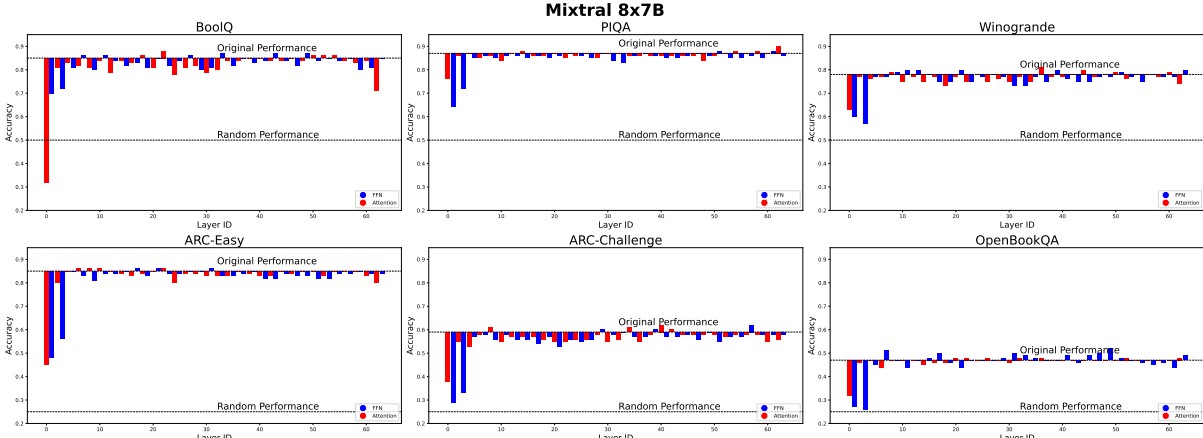

Figure 5: **Layer ablation result of Mixtral 8x7B**. X-axis shows the layer ID of the removed layer. Y-axis shows the accuracy after this layer is removed. Attention layers are colored in red, while MoE layers are colored in blue. Removing a single layer generally causes a decrease in model performance. However, even after ablating cornerstone layers, the performance of Mixtral 8x7B remains above random guessing, suggesting a more balanced contribution among the layers for LLMs with MoE layers instead of FFN layers. More discussion in Section 6.3.

ideas, as doing so will be crucial in advancing our understanding of layer-specific roles in LLMs.

## 7 Conclusion and Future Work

In this study, we investigated the significance and contribution of individual layers in LLMs using Shapley values and layer ablation. Our results based on Shapley values revealed that certain layers, typically early in the model, exhibit a dominant contribution to the model performance, which we term as cornerstone layers. Layer ablation experiments demonstrated that removing a single cornerstone layer can cause the model to collapse and perform random guessing, highlighting their critical role. Conversely, removing other non-cornerstone layers resulted in marginal performance changes, indicating redundancy in the model architecture.

Future works can continue the study on layer importance in groups of layers instead of one single layer. Investigating the specific reasons behind the importance of cornerstone layers could provide deeper insights into LLM functionality and inspire newer LLM structures that promote model transparency, remove redundant parts, and improve inference efficiency.

## 8 Limitation

Our sampling method for estimating Shapley values may introduce bias, potentially affecting the accuracy of our layer importance estimations. In addition, our analysis focuses on the general contribution of individual layers without examining how

exactly different layers interact with each other and incorporate information from other layers. Future work on layer interaction can also help validate our Hypothesis 2. Furthermore, a deeper exploration into the unique functions of the early layers remains an open avenue for future research. Understanding why these layers play a critical role could provide valuable insights into optimizing model performance. Future work on layer functionalities can help validate our Hypothesis 1.

## 9 Ethical Consideration

Transparency and explainability are key in deploying LLMs, especially in sensitive applications like healthcare or legal systems. Understanding the role of cornerstone layers can enhance explainability, but it is essential to communicate these findings clearly to non-expert stakeholders to foster trust and accountability. In addition, the identification of cornerstone layers and their critical roles may lead to more targeted and efficient model optimization. However, it is crucial to ensure that these optimizations do not inadvertently introduce biases or reinforce existing ones. Lastly, the redundancy identified in other layers suggests the potential for model simplification, which could reduce computational costs and environmental impact. However, such reductions must balance performance and fairness, ensuring that simplified models do not compromise on ethical standards.

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

## A  Estimated Shapley Value

This section provides an additional plot to show the actual value of estimated Shapley values. Figure 6 illustrates the bar plot of Shapley values across different layers in the model. The plot reveals the precise contributions of each layer, allowing interested readers for further reference.

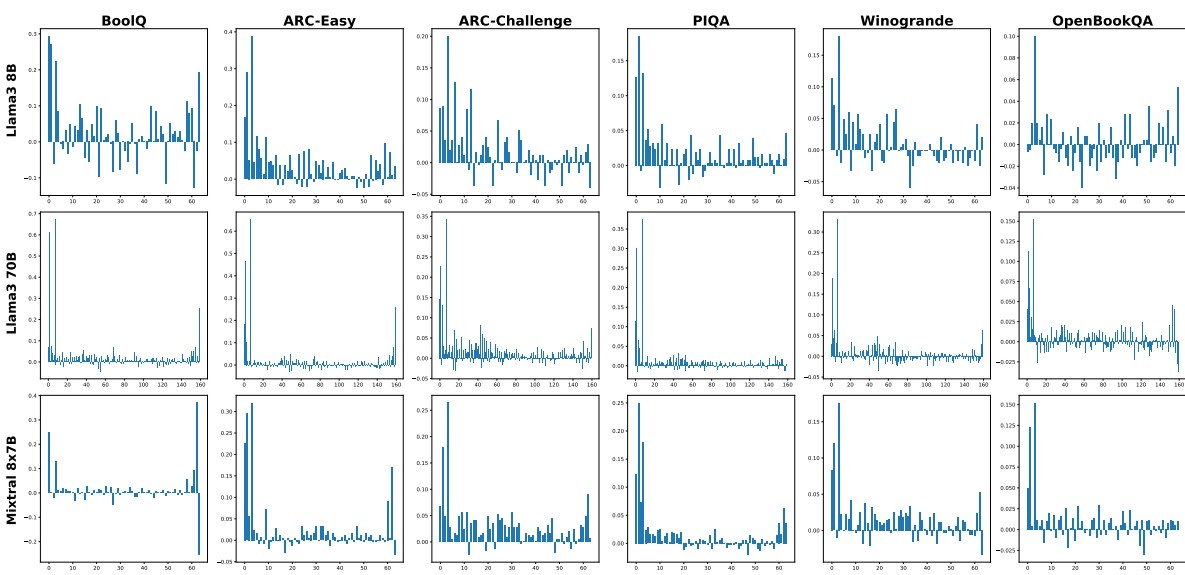

Figure 6: Estimated Shapley value result. X-axis shows the layer ID of the layer. Y-axis shows the estimated Shapley value.