# OpenReview forum: "Investigating Layer Importance in Large Language Models"
_EMNLP/2024/Workshop/BlackBoxNLP — BlackboxNLP 2024_

### Official Review · Reviewer_E8YY · 2024-09-05

**Overall Assessment:** 4
**Confidence:** 4

**Best Paper:**

1

**Best Paper Justification:**

-

**Comments Questions Suggestions And Typos:**

Typos:
- ref spacing line 036
- ref spacing line 155
- I would say that "the performance of the model collapses", not the model itself. Lines 363 and 402.
- Titles and subtitles: I would change "Analyse" to e.g. "Analysing" or "Analysis (of...)". Line 104.
- Same goes for line 133 (Study), 191 (Estimate).
- "Experiment setup" -> "Setup" or "Experimental setup". Line 263.

Figures:
- Too small, especially 3, 4, 5. Especially the texts and legends. You could maybe pick one or two tasks and show only them for the three models. Leave the plots for the remaining tasks in the appendix. I really like figure 2, but try to increase the font.

Title:
- I would personally add something to the paper title, e.g. the methods used or your findings, or the concept "cornerstone layers". Otherwise it sounds a bit generic.

**Paper Summary:**

The paper investigates the importance of individual layers in LLMs, combining the results given by two different methods: Shapley-based interpretation and layer ablation. They find that certain layers are more important than others through the first approach, and that by carefully removing these layers through the second approach, the performance of the models in fact drastically decreases. This doesn't happen for layers that are not 'cornerstone layers'.

**Summary Of Strengths:**

The idea of the paper is very clear and the execution is well done. The paper is well-structured. The authors do not only test their hypothesis with one method (which may be unfaithful) but show/confirm how a different method confirms the validity of the former.

**Summary Of Weaknesses:**

- It wasn't clear for me what the difference is between the authors' work and Ghorbani an Zou (2020) in terms of shapley value and layer ablation results, under "Early truncation" (potential novelty issue).
- Lines 309, 339, 343. I have the impression that we would need a baseline to tell how "disproportionate" these numbers are
- Line 226: why removing maximally 4 layers? I suggest to explain it better. Would a different n layers affect the conclusions in your work?
- I wonder if it isn't just "logical" that initial cornerstone layers are primarily responsible for processing initial input embeddings (line 406), since that's where the input reaches the model?

---

### Official Review · Reviewer_2fpe · 2024-09-08

**Overall Assessment:** 4
**Confidence:** 3

**Best Paper:**

1

**Best Paper Justification:**

NA

**Comments Questions Suggestions And Typos:**

- You should define what ‘mechanistic’ means in section 5.
- I don’t think I understand the authors point about the complementary nature of Shapley values and layer ablations. Could you provide concrete examples? I am not sure I understand how Shapley value provide insights into layer interactions.
- What could be the reason for higher percentage of Shapley values for top-3 layers in Llama3-70B (37%) vs Llama3-8B (29%)?
- How do you determine a cornerstone layer? From Figure 2, Attn 0 doesn’t have a significant impact on ARC-Challenge and OpenBookQA. But, Table 2 identifies it as a cornerstone layer for Llama3-8B. Ablation results in Figure 3 support the importance Attn 0.
- Can you report the correlation between layer importance identified by the two proposed methods, Shapley values and performance ablation?
- In their limitations section, authors mention that their methodology doesn’t account for layer interaction (lines 453-458). However, the text in the main paper states the contrary (lines 256-261). This is related to my previous question about the author’s claim about the complementary nature of Shapley values and layer ablations.
- There is some similarity to the cited work Gromov et al., 2024. It would be useful to provide a more detailed comparison in the paper.

**Paper Summary:**

This paper studies two interpretability techniques for layer importance in Llama-style transformers, Shapley values and layer dropping. Specifically, on the reading comprehension task, the authors find specific low-level layers that provide the most benefit on downstream task performance. The results are consistent across model sizes and downstream tasks.

The paper could benefit from some editing and additional experiments on generative tasks. See my comments in weaknesses and questions to the authors.

**Summary Of Strengths:**

- The empirical results in this paper are quite interesting and provides valuable directions for future work on reading comprehension tasks with Llama-style models. Instead of just scaling the number of layers in the transformer, future work could focus on improve the effectiveness of each layer.
- The two interpretability approaches studied in this paper are well-motivated and insightful at drawing conclusions. Shapley values has been used for data valuation before, but its interesting to see it applied to study layer importance. The layer ablation method is simple and mostly provides support to the results from Shapley value analysis.
- The results are consistent across the six reading comprehension benchmarks, validating the proposed methodology.

**Summary Of Weaknesses:**

- The analysis in this paper is limited to reading comprehension datasets. For this task, the importance of initial layers is expected. It would be useful to extend the analysis to generative tasks such as text summarization.
- Authors make a claim about the complementary nature of their proposed methods, Shapley values and layer ablations. They don’t provide enough evidence in the paper to support this claim. See my question below about the claims on layer interaction.
- The definition for a cornerstone layer could be clearer. I couldn’t find a concrete definition in the text (also see my question below).
- The exact computation of Shapley values is not clear from the text of the paper. The authors provide a high-level description, but it would be useful to provide the full algorithm or pseudocode for it.

---

### Official Review · Reviewer_X5JT · 2024-09-09

**Overall Assessment:** 2
**Confidence:** 4

**Best Paper:**

1

**Best Paper Justification:**

.

**Comments Questions Suggestions And Typos:**

# Observations
- Row 210: "[..] both v(S ∪ {i}) and v(S) are essentially random guesses. To exploit this, we limit our sampling to scenarios where layers are removed up to a certain level". This is unclear: why not remove instead if both are low, e.g., with a ridge filter?
- " Each layer primarily influences its immediate subsequent layers and is influenced by its immediate preceding layers." By definition, it influences all the following ones, not just its following. This is particularly important because players in the game are inherently dependent on each other, thus they are strongly dependent, thus the assumption of players independence does not really hold. This is also strongly suggested by the results, which assign high importance to early layers.
- Following the previous point, I'd argue a random perturbation-based approach would be more proper.
- A simple baseline would be the norm of the weights of the layer, which provide, by design, a quantitative indication of the numerical impact. Same holds for their activations.
- As mentioned in the literature background, several works have identified some heads as more relevant than others? Does this affect layer importance computation?
- Personally, I find that a more interesting comparison would be a mirror of Table 4, but on cornerstone layers. You could make some space by removing Fig 3., as it shows similar findings to Fig

# Soundness
2/5.

**Paper Summary:**

In this manuscript, the authors propose a study of layer importance of Large Language Models (LLMs).

**Summary Of Strengths:**

# Strong points
1. Experimentation on modern models, and varied dataset types.
2. Analysis transversal to the current ones in the literature

**Summary Of Weaknesses:**

# Weak points
1. Deeply flawed assumption and experimental design.

---

### Decision · Program_Chairs · 2024-09-20

**Decision:**

Accept

**Comment:**

This paper investigates the importance of layers in LLMs using two interpretation techniques—Shapley values and layer ablation. The empirical results are interesting and insightful. While one reviewer raised concerns about the assumption made in the paper, this alone is not a strong reason for rejection. I recommend that the authors address this concern and other comments in the camera-ready version.